# Prosthesis–Patient Mismatch in Small Aortic Annuli: Self-Expandable vs. Balloon-Expandable Transcatheter Aortic Valve Replacement

**DOI:** 10.3390/jcm11071959

**Published:** 2022-04-01

**Authors:** Jerome Ferrara, Alexis Theron, Alizee Porto, Pierre Morera, Paul Luporsi, Nicolas Jaussaud, Vlad Gariboldi, Frederic Collart, Thomas Cuisset, Pierre Deharo

**Affiliations:** 1Département de Cardiologie, CHU Timone, 13005 Marseille, France; thomas.cuisset@ap-hm.fr (T.C.); pierre.deharo@ap-hm.fr (P.D.); 2Département de Chirurgie Cardiaque, CHU Timone, 13005 Marseille, France; alexis.theron@ap-hm.fr (A.T.); alizee.porto@ap-hm.fr (A.P.); pierre.morera@ap-hm.fr (P.M.); nicolas.jaussaud@ap-hm.fr (N.J.); vlad.gariboldi@ap-hm.fr (V.G.); frederic.collart@ap-hm.fr (F.C.); 3Département de Cardiologie, Centre Hospitalier Bastia, 20600 Bastia, France; paulluporsi@yahoo.fr; 4Inserm, Inra, C2VN, Aix-Marseille Université, 13005 Marseille, France; 5Faculté de Médecine, Aix-Marseille Université, 13005 Marseille, France

**Keywords:** TAVR, mismatch, small

## Abstract

Prosthesis–patient mismatch (PPM) is associated with worse outcomes following surgical aortic valve replacement (SAVR). PPM has been identified in a significant proportion of TAVR, particularly in patients with small aortic annuli. Our objective was to evaluate the hemodynamic performances of balloon-expandable (BE) (Sapiens 3^TM^) versus two different self-expandable (SE) (Evolut Pro^TM^, Accurate Neo^TM^) TAVR devices in patients with small aortic annulus defined by a computed tomography aortic annulus area (AAA) between 330 and 440 mm^2^. We enrolled 131 consecutive patients corresponding to 76 Sapiens 3 23 mm (58.0%), 26 Evolut Pro (19.9%) and 29 Accurate Neo (22.1%). Mean age was 82.5 ± 7.06 years, 22.9% of patients were male and mean Euroscore was 4.0%. Mean AAA was 374 ± 27 mm^2^ for Sapiens 3, 383 ± 29 mm^2^ for Corevalve Evolut Pro and 389 ± 25 mm^2^ for Accurate Neo. BE devices were associated with significantly higher rates of PPM (39.5%) as compared to SE devices (15.4% for Corevalve Evolut Pro and 6.9% for Accurate Neo) (*p* < 0.0001). Paravalvular leaks ≥ 2/4 were more often observed in SE devices (15.4% for Corevalve Evolut Pro and 17.2% for Accurate Neo) than in BE devices (2.6%) (*p* = 0.007). In conclusion, SE TAVR devices did achieve better hemodynamic results despite higher rates of paravalvular leaks. Therefore, SE TAVI devices could be considered as first choice in small aortic anatomy.

## 1. Introduction

Continuous development has improved the results of transcatheter aortic valve replacement (TAVR). Eligible surgical risk classes and volume of TAVR have expanded over years, and this technique has exceeded surgical aortic valve replacement (SAVR) in the US and European countries [1]. While prosthesis–patient mismatch (PPM) has been associated with impairment of long term survival following SAVR [2,3,4,5,6,7], particular attention is required for TAVR, considering that this technique became the first line aortic valve replacement strategy. In large cohorts, despite a lower incidence than in SAVR, PPM has been identified in a significant proportion of TAVR and particularly in patients with small aortic annuli [8,9]. The balloon-expandable (BE) valve and the self-expandable (SE) valve have been proven to be effective in the management of severe symptomatic aortic stenosis with some difference in hemodynamic results [10,11,12]. In this population of patients with small aortic annuli, valvular anatomic characteristics should perhaps dictate the type of valve to be used.

Therefore, our objective was to evaluate the hemodynamic performances of balloon-expandable (BE) versus two different self-expandable (SE) TAVR devices in patients with small aortic annulus defined by a computed tomography (CT) aortic annulus area between 330 and 440 mm^2^.

## 2. Materials and Methods

### 2.1. Study Population and Design

In La Timone Hospital, from June 2019 to April 2021, we prospectively enrolled all patients with a small aortic annulus between 330 and 440 mm^2^ on cardiac CT undergoing TAVI for severe symptomatic aortic stenosis. TAVR was performed using either a 23-mm Sapiens 3TM (Edwards Lifesciences, Irvine, CA, USA) balloon-expandable (BE) valve, an Accurate NeoTM S or M (Boston Scientific, Marlborough, MA, USA) or a 26- or 29-mm Evolut ProTM (Medtronic, Dublin, Ireland) self-expanding (SE) valve, according to physician preference. This study complied with the provisions of the Declaration of Helsinki and all patients provided written consent. All patients were over 18. The exclusion criteria were patients who did not receive a complete post-procedure echocardiography evaluation because of intraoperative or early postoperative death; incomplete ultrasound data not allowing calculation of the effective orifice area (EOA) of the prosthesis; and missing echocardiography data at 1 month of follow-up. As the patients were treated with commercially available devices (validated in those indications) we did not require an ethical committee for implantation of BE in patients with small aortic annuli. Of note, this study complied with the provisions of the Declaration of Helsinki and all patients provided written consent.

### 2.2. Procedure

Each patient underwent a multidisciplinary preoperative heart team evaluation combined with a CT scan to validate the indication for the procedure. Transfemoral access was used almost exclusively.

### 2.3. PPM Definition

According to the Valve Academic Research Consortium (VARC)-3 criteria [13], we defined the severity of PPM according to the indexed EOA (iEOA) of the prosthetic valve and classified PPM severity as: None or mild, >0.85 cm^2^/m^2^; moderate, between 0.85 and 0.65 cm^2^/m^2^; and severe, <0.65 cm^2^/m^2^.

### 2.4. Paravalvular Leak Definition

Aortic regurgitation (paravalvular leak) was assessed by using colourflow Doppler signal and graded in 5 groups: None or trivial (=0/4), mild (=1/4), mild-to-moderate (=2/4), moderate-to-severe (=3/4), or severe (=4/4).

### 2.5. Small Aortic Annulus Definition

We took as definition for small annuli the patients who could benefit from the smallest Sapiens 3 valve available in France (i.e., Sapiens 3 23 mm valve) according to the manufacturers’ labeling ranking (330–440 mm^2^).

### 2.6. Follow Up

Follow up was done at one month with clinical consultation and echocardiography. During clinical consultation, vital status and NYHA status were checked. We evaluated left ventricular ejection fraction (LVEF), effective orifice area (EOA), indexed EOA, PPM and grade of paravalvular leak.

### 2.7. Endpoints

The primary study endpoint was the occurrence of moderate or severe PPM at one month. The secondary endpoints were the occurrence of paravalvular leak ≥ 2 at one month and pacemaker implantation during the thirty-first days.

### 2.8. Statistical Analysis

Statistical analysis was performed using PASW Statistics version 18.0. Continuous variables were reported as means and standard deviation or as medians and range (according to their distribution), and categorical variables were reported as count and percentages. Standard two-sided tests were used to compare continuous characteristics (Student *t* or Mann–Whitney *U* tests) or categorical characteristics (chi-square or Fisher exact tests) among patient groups. For all tests, statistical significance was defined as *p* < 0.05.

## 3. Results

A total of 143 consecutive patients were included prospectively from between June 2019 and April 2021. Twelve patients were excluded (Figure 1). We finally studied 131 patients corresponding to 76 Sapiens 3^TM^ 23 mm (58.0%), 26 Evolut Pro^TM^ (19.9%) and 29 Accurate Neo^TM^ (22.1%). Mean age was 82.5 ± 7.06 years, 22.9% of patients were male and mean Euroscore was 3.9%. Transfemoral TAVR was performed in 99.2% of cases. There was no difference in baseline characteristics between groups (Table 1, Table 2 and Table 3). Mean AAA was non significantly different within the three groups (374 ± 27 mm^2^ for Sapiens 3383 ± 29 mm^2^ for Corevalve Evolut Pro and 389 ± 25 mm^2^ for Accurate Neo; *p* = 0.06). Postdilatation was performed in seven patients (zero Sapiens 3^TM^, four Accurate Neo^TM^ and three Evolut Pro^TM^).

### 3.1. Occurrence of Moderate or Severe PPM at One Month

Primary endpoint occurred for 30/76 patients (39.5%) in the Sapiens 3^TM^ 23 mm group, 4/26 patients (15.4%) in the Evolut Pro^TM^ and 2/29 patients (6.9%) in the Accurate Neo^TM^ group with a significant difference between the three groups (*p* < 0.0001). Interestingly, BE devices were associated with higher rates of moderate PPM (26.3%) with a significant difference (*p* = 0.008) and severe PPM (13.2%) without significant difference as compared to SE (Figure 2 and Figure 3).

Furthermore, mean gradients were higher in the Sapiens 3^TM^ 23 mm group (13.9 ± 4.81 mmHg) than in the Evolut Pro^TM^ group (9.2 ± 3.92 mmHg) and in the Accurate Neo^TM^ group (9.3 ± 3.42 mmHg) with a significant difference (*p* < 0.0001).

### 3.2. Occurrence of Paravalvular Leak at One Month and Pacemaker Implantation during the Thirty-First Days

Paravalvular leaks ≥ 2/4 were more often observed in the Evolut Pro^TM^ group (4/26; 15.4%) and in the Accurate Neo^TM^ group (5/17; 17.2%) than in the Sapiens 3^TM^ 23 mm group (2/76; 2.6%) with a significant difference between the three groups (*p* < 0.007) (Figure 2).

Pacemaker implantation during the thirty-first days was the same in the three group (Sapiens 3^TM^ 23 mm 18.4%, Evolut Pro^TM^ 15.4%, Accurate Neo^TM^ 17.2%) (*p* = 0.93) (Table 4).

## 4. Discussion

The main findings of this study are:(1)PPM was more often observed with BE TAVR than with SE TAVR and mean gradients were higher with BE TAVR(2)The majority of paravalvular leaks ≥ 2/4 occurred with SE TAVR compared to BE TAVR

PPM remains an under-explored complication of TAVR, which may be associated with worse clinical outcomes and accelerated structural valve deterioration [14]. Small aortic annuli are at particular risk of PPM, mostly when treated with intra-annular and bulkier devices, due to higher risk of overcrowding of the left ventricle outflow tract [15]. When considering BE TAVR, 23 mm size of the Sapiens 3 valve has been associated with significant incidence of PPM compared to larger sizes [8]. We confirmed that supra-annular TAVR design achieved larger iEOA and better hemodynamics in small aortic annuli, although our data are in favour of overall low risk of PPM. Our results corroborate those from the CHOICE randomized trial, with a more accurate definition of small aortic annulus based on CT and addition of a second SE TAVR device [16]. Hase et al. compared Sapiens 3 and Corevalve and showed that SE TAVR had better hemodynamic performance in small annuli [17]. Recently, studies have questioned the prognostic impact of mismatch in the global population of TAVR. Concerning SE valves, Tang et al. showed that the presence of a severe mismatch did not impact one-year mortality [18]. Conversely, for this type of valve, Leone et al. demonstrated that severe PPM had an impact on the prognosis of patients after TAVR (higher rates of all cause mortality) [19]. However, in light of our results, patients with a small aortic annulus seem to benefit from extensive screening. It will probably participate in the reduction of the occurrence of a mismatch and structural valve deterioration.

Another important finding of our study is the presence of a higher rate of paravalvular leaks with SE valves compared to the two BE TAVR. This finding is supported by the SOLVE TAVI study [12]. In these small anatomies, careful CT analysis could identify high-risk situations for PVL (aortic annuli with important calcifications).

### Strength and Limitation

There were several limitations to this study, the most important one being its single center nature with a small study population.

Then, the echocardiography measurements, particularly the LVOT diameter and LVOT VTI assessment, are particularly challenging and may overestimate the incidence of PPM [20]. The use of predicted iEOA could allow to get rid of these echocardiography variabilities [21].

Moreover, our assessment of paravalvular leak in four grades may increase their severity compared to a definition based on five grades [13] which would explain the higher rates of paravalvular leaks in our study.

## 5. Conclusions

In small aortic annuli, SE TAVR devices did achieve better hemodynamic results despite higher rates of paravalvular leaks. Our work suggests that, in this population, SE TAVR design may be favoured. Indeed, there is a compromise to make in small aortic annuli between PVL and PPM and a special interest should be given to the choice of valve.

## Figures and Tables

**Figure 1 jcm-11-01959-f001:**
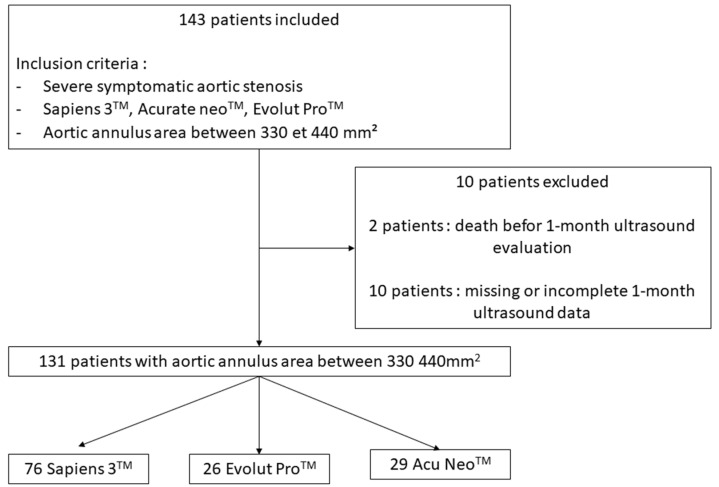
Flow chart.

**Figure 2 jcm-11-01959-f002:**
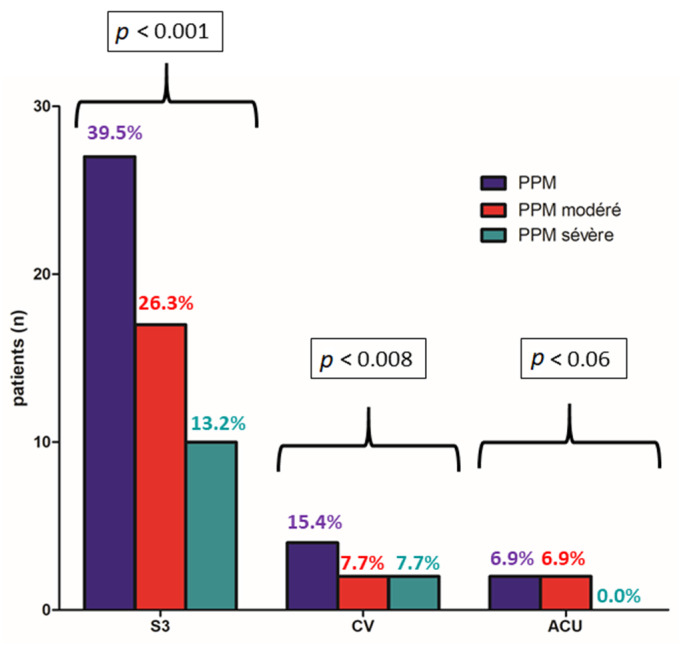
Occurrence of moderate or severe PPM at one month according to balloon-expandable (Sapiens 3^TM^) vs. self-expandable (Evolut Pro^TM^, Accurate Neo^TM^) TAVR devices.

**Figure 3 jcm-11-01959-f003:**
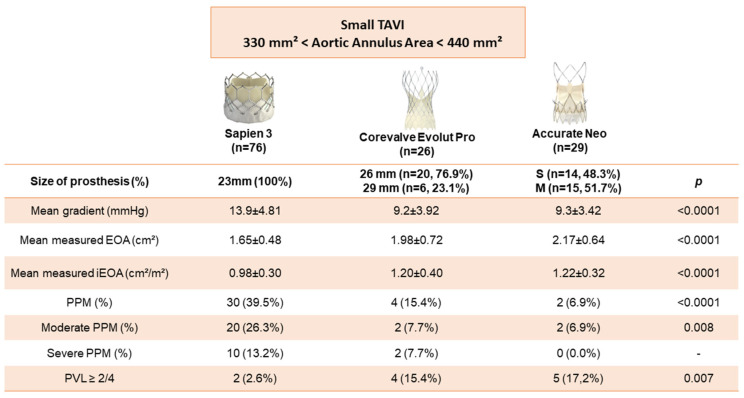
Hemodynamic characteristics at 1 month after TAVR according to balloon-expandable (Sapiens 3^TM^) vs. self-expandable (Evolut Pro^TM^, Accurate Neo^TM^) TAVR devices.

**Table 1 jcm-11-01959-t001:** Patients’ characteristics at the baseline.

Patient Characteristics at the Baseline	General Population *n* = 131	Sapiens 3^TM^ *n* = 76	Evolut Pro^TM^ *n* = 26	Accurate Neo^TM^ *n* = 29	*p* Value
Age-yr	82.5 ± 7.06	82.45 ± 7.28	80.35 ± 6.96	84.59 ± 6.10	0.52
Male sex–no (%)	30/131 (22.90%)	16/76 (21.05%)	6/26 (23.08%)	8/29 (27.59%)	0.78
Body surface–m^2^	1.74 ± 0.25	1.76 ± 0.31	1.68 ± 0.17	1.77 ± 0.20	0.63
BMI-kg/m^2^	25.73 ± 4.76	25.24 ± 4.33	25.83 ± 4.07	26.93 ± 6.15	0.26
EuroSCORE II	3.92 ± 2.99	4.17 ± 3.05	3.52 ± 3.87	3.59 ± 1.73	0.51
Diabetes mellitus-no./total no. (%)	28/131 (21.37%)	17/76 (22.37%)	5/26 (19.23%)	6/29 (20.69%)	0.94
Hypertension-no./total no. (%)	91/131 (69.47%)	55/76 (72.37%)	17/26 (65.38%)	19/29 (65.52%)	0.70
Dyslipidemia-no./total no. (%)	68/131 (51.91%)	42/76 (55.26%)	12/26 (46.15%)	14/29 (48.28%)	0.79
Dialysis-no./total no. (%)	4/131 (3.05%)	2/76 (2.63%)	1/26 (3.85%)	1/29 (3.45%)	0.94
COPD-no./total no. (%)	17/131 (12.98%)	10/76 (13.16%)	2/26 (7.69%)	5/29 (17.24%)	0.57
Smoke-no./total no. (%)	42/131 (32.06%)	25/76 (32.89%)	7/26 (26.92%)	10/29 (34.48%)	0.81
CAD-no./total no. (%)	51/131 (38.93%)	34/76 (44.74%)	7/26 (26.92%)	10/29 (34.48%)	0.27
Atrial fibrillation-no./total no. (%)	46/131 (35.11%)	28/76 (36.84%)	8/26 (30.77%)	10/29 (34.48%)	0.85
Cancer-no./total no. (%)	23/131 (17.56%)	13/76 (17.11%)	4/26 (15.38%)	6/29 (20.69%)	0.86
NYHA:					0.31
- 2 - 3 - 4	54/131 (41.22%) 63/131 (48.09%) 14/131 (10.69%)	34/76 (44.74%) 32/76 (42.10%) 10/76 (13.16%)	7/26 (26.92%) 17/26 (65.38%) 2/26 (7.69%)	13/29 (44.83%) 14/29 (48.28%) 2/29 (6.90%)	

BMI: Body Mass Index; COPD: Chronic Obstructive Pulmonary Disease; CAD: Coronary artery Disease.

**Table 2 jcm-11-01959-t002:** Echocardiography data at the baseline and access data.

Echocardiography Data at the Baseline and Access Data	General Population *n* = 131	Sapiens 3^TM^ *n* = 76	Evolut Pro^TM^ *n* = 26	Accurate Neo^TM^ *n* = 29	*p* Value
LVEF-no./total no. (%)	60.11 ± 11.82	60.83 ± 10.03	60.15 ± 10.74	63.28 ± 9.35	0.45
Mean trans-aortic gradient-mmHg	54.27 ± 15.37	55.18 ± 17.58	52.62 ± 11.75	53.38 ± 11.83	0.72
EOA-cm^2^	0.68 ± 0.17	0.68 ± 0.17	0.68 ± 0.19	0.71 ± 0.17	0.66
iEAO–cm^2^/m^2^	0.40 ± 0.11	0.40 ± 0.11	0.41 ± 0.10	0.40 ± 0.09	0.82
Aortic annulus area-no./total no. (%)	378.69 ± 27.45	373.76 ± 27.30	383.14 ± 29.00	383.76 ± 25.52	0.06
Transfemoral access-no./total no. (%)	128/131 (99.2%)	75/76 (98.68%)	26/26 (100%)	29/29 (100%)	0.70

LVEF: Left Ventricular Ejection Fraction; EOA: Effective Orifice Area; iEAO: indexed Effective Orifice Area.

**Table 3 jcm-11-01959-t003:** Echocardiography data at one month.

Echocardiography Data at One Month	General Population *n* = 131	Sapiens 3^TM^ *n* = 76	Evolut Pro^TM^ *n* = 26	Accurate Neo^TM^ *n* = 29	*p* Value
LVEF-%	63.30 ± 8.06	62.78 ± 7.90	63.77 ± 10.16	64.24 ± 6.33	0.67
Mean trans-aortic gradient-mmHg	11.98 ± 4.92	13.92 ± 4.81	9.20 ± 3.92	9.31 ± 3.42	<0.001
EOA-cm^2^	1.83 ± 0.60	1.65 ± 0.48	1.98 ± 0.72	2.17 ± 0.64	<0.001
iEAO–cm^2^/m^2^	1.08 ± 0.35	0.98 ± 0.30	1.20 ± 0.40	1.22 ± 0.32	<0.001
PPM-no./total no. (%)	36/131 (27.48%)	30/76 (39.5%)	4/26 (15.4%)	2/29 (6.9%)	<0.001
Moderate PPM-no./total no. (%)	24/131 (18.32%)	20/76 (26.3%)	2/26 (7.7%)	2/29 (6.9%)	0.008
Severe PPM-no./total no. (%)	12/131 (9.16%)	10/76 (13.2%)	2/26 (7.7%)	0/29 (0.0%)	0.06
Paravalvular leak ≥ 2-no./total no. (%)	11/131 (8.40%)	2/76 (2.6%)	4/26 (15.4%)	5/29 (17.2%)	0.007

LVEF: Left Ventricular Ejection Fraction; EOA: Effective Orifice Area; iEAO: Indexed Effective Orifice Area; PPM: Prosthesis–Patient Mismatch.

**Table 4 jcm-11-01959-t004:** Post-procedural complication.

Post Procedural Complication	General Population *n* = 131	Sapiens 3^TM^ *n* = 76	Evolut Pro^TM^ *n* = 26	Accurate Neo^TM^ *n* = 29	*p* Value
Pacemaker-no./total no. (%)	23/131 (17.56%)	14/76 (18.42%)	4/26 (15.38%)	5/29 (17.2%)	0.93
Vascular complication-no./total no. (%)	11/131 (8.40%)	8/76 (10.5%)	2/26 (7.69%)	1/29 (3.4%)	-
Bleeding complication-no./total no. (%) - type 1 - type 2	9/131 (6.87%) 6/131 (4.58%) 2/131 (1.53%)	6/76 (7.89%) 6/76 (7.89%) 0/76 (0.00%)	2/26 (7.69%) 0/26 (0.00%) 2/26 (7.69%)	1/29 (3.4%) 0/29 (0.00%) 1/29 (3.4%)	-

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
