# Peer review of "Prosthesis–Patient Mismatch in Small Aortic Annuli: Self-Expandable vs. Balloon-Expandable Transcatheter Aortic Valve Replacement"

_jcm, 2022, doi:10.3390/jcm11071959_

Round 1

Reviewer 1 Report

I read with interest this single center study aimed at evaluating the hemodynamic per formances of BE (Sapiens 3TM) in 76 patients versus two different SE  (Evolut Pro ™, Acurate neo TM) in 26 and 29 patients respectively TAVR devices in patients with small aortic annulus defined by 330<AAA<440mm2. (age  82.5±7.06 years, 22.9% male , mean Euroscore was 4.0 %). BE devices were associated with significantly higher rates of PPM 33 (39.5%) as compared to SE devices (15.4% for Corevalve Evolut Pro and 6.9% for Accurate neo) 34 (p<0.0001). Paravalvular leaks≥2/4 were more often observed in SE devices (15.4% for Corevalve 35 Evolut Pro and 17.2% for Accurate Neo) than in BE devices (2.6%) (p=0.007). The Authors conclude stating that SE TAVR devices did achieve better hemodynamic results despite higher rates of paravalvular leaks. Therefore, SE TAVI devices could be considered as first choice in small aortic anatomy. 

This is an interesting study whose limitation has been clearly and honestly declared by the Authors. My concerns regards:

  1. Study design: it is not celar to me whether these patients underwent TAVR prospectively and then retrospectively analyzed, or if patients were prospectively enrolled purposely for this study. If the latter is the case, the choice of BE should have been motivated and an ethical committee should have cleared it. Please, clarify
  2. the definition of small AAA should have a reference
  3. echocardiography: was it performed/read by one single investigator or not? Please, clarify. How was para valvular leak quantified? please specify and provide specific references
  4. Recently a number of studies and meta analysis explored this topic. These should be more accurately quoted and discussed
  5. The Novelty of these findings should be underscored if any, and the adjunctive value of the paper more clearly stated 

Author Response

  1. Study design: it is not celar to me whether these patients underwent TAVR prospectively and then retrospectively analyzed, or if patients were prospectively enrolled purposely for this study. If the latter is the case, the choice of BE should have been motivated and an ethical committee should have cleared it. Please, clarify

Patients were prospectively enrolled in our analysis. As the patients were treated with commercially available devices (validated in those indications) we did not require an ethical committee for implantation of BEV in patients with small aortic annuli. Indeed BEV are validated in annuli from 330mm2 to 440mm2. Of note in the recent PARTNER 3 study the 23mm valve size (annuli from 330mm2 to 440mm2) was implanted in 30% of the patients (Mack MJ, et al; PARTNER 3 Investigators. Transcatheter Aortic-Valve Replacement with a Balloon-Expandable Valve in Low-Risk Patients. N Engl J Med. 2019 May 2;380(18):1695-1705. doi: 10.1056/NEJMoa1814052. Epub 2019 Mar 16. PMID: 30883058.). Therefore, we did not involve an ethical committee for this specific analysis. Of note this study complied with the provisions of the Declaration of Helsinki and all patients provided written consent. The choice of prosthesis implanted in those patients was left to the discretion of the operators according to available TAVI prosthetic valve (Sapien Acurate and Core valve) and native aortic valve anatomy.

We now clarified this in our paper.

  1. The definition of small AAA should have a reference

Thank you for your intersting comment. Indeed, to date, no clear consensus has been established regarding the cutoff value for defining SAA, which results in multiple definitions used in different studies for the same concept.

We took as definition for small annuli, the patients who could benefit from the smallest Sapiens 3 valve available in france (i.e. Sapien 3 23mm valve) according to the manufacturers’ labeling ranking (330-440mm2).

Freitas-Ferraz et al suggested as definition an aortic annulus mean diameter≤23mm and that is close to the commercial data (Freitas Ferraz, et al, Aortic stenosis and small aortic annulus. Circulation; 3 Jun 2019; 139:2685–2702 ; PMID: 31157994 ; DOI: 10.1161/CIRCULATIONAHA.118.038408)

However, the fact that the manufacturers’ labeling of valves is not uniform and refers inconsistently to the diameter of the external sewing ring (mechanical prosthesis), mounting ring (stented bioprostheses), or internal orifice (allografts and some stentless bioprostheses), may lead to erroneous conclusions regarding the comparison of hemodynamic performance between valves.

On a specific TAVI perspective, it seems that the current challenges regarding small aortic annuli focus on the smallest valve available, which is the 23mm in case of BEV. Indeed, to compare in those small aortic anuli BEV versus self expanding valve (SEV) we had to select small annuli according to Sapiens 3 23mm manufacturers’ labeling ranking

  1. Echocardiography: was it performed/read by one single investigator or not? Please, clarify. How was para valvular leak quantified? please specify and provide specific references

Thanks for your comment. We now provide detail regarding classification in the material and method section.

Postprocedural TTE was intended to be performed at day 2 after the procedure and was performed, at the latest, before hospital discharge and at day 30 by experienced cardiologists blinded. Mitral and aortic regurgitation (AR) (paravalvular leak (PVL)) were assessed by using a colorflow doppler signal and graded in 5 groups as none or trivial (=0/4), mild (=1/4), mild-to-moderate (=2/4), moderate-to-severe (=3/4), or severe (=4/4). Native aortic regurgitation and aortic paravalvular leak were evaluated according to the European Association of Echocardiography guidelines  and the American Society of Echocardiography recommendations  by the use of a multiparametric and integrative approach rather than a single measurement. In case of post-TAVR ARs, because they are often paravalvular, the evaluation relied more heavily on the circumferential extent of the paravalvular jet(s) as evaluated just below the bioprosthesis on the short-axis view, than on the other parameters. None, mild, mild-to-moderate, moderate-to-severe, and severe post-TAVRs were defined according to American Society of Echocardiography guidelines (Zoghbi et al. Guidelines for the Evaluation of Valvular Regurgitation After Percutaneous Valve Repair or Replacement. Journal of the American Society of Echocardiography ; 2019 Feb 20 ; PMID: 30797660DOI: 10.1016/j.echo.2019.01.003) with the following adaptation that, similar to the European Association of Echocardiography proposal for the evaluation of ARs of the native valves ( Lancellotti P, et al. European Association of Echocardiography recommendations for the assessment of valvular regurgitation. Part 1: aortic and pulmonary regurgitation (native valve disease). European Heart Journal - Cardiovascular Imaging.

2010;11:223–244. ; Pibarot P, et al. Assessment of Paravalvular Regurgitation Following TAVR. JACC: Cardiovascular Imaging. 2015;8:340–360), moderate post-TAVR PVLs were subdivided in mild-to-moderate and moderate-to-severe PVLs. When several PVL were present, PVL was expressed as an overall grade unless otherwise stated. A valvular regurgitation>= 2 was considered significant. Pre stent LVOT diameter was measured according to recommendation. The Effective Orifice Area (EOA) was calculated according to the continuity equation. The indexed EOA (iEOA) was calculated as the EOA divided by the body surface area (BSA). Moderate PPM was defined by an 0.65≤iEOA≤0.85 cm2/m2 (0.55≤iEOA≤0.70 cm2/m2 if BMI ≥30 kg/m2) and severe PPM was defined by an iEAO<0.65 cm2/m2.(≤0,55 if BMI ≥30 kg/m2).

  1. Recently a number of studies and meta analysis explored this topic. These should be more accurately quoted and discussed

Following your comment, we did perform a new review of the litterature regarding small aortic annuli and TAVI.

Recently, Leone et al (JACC Cardiovascular Intervention 2021) published a paper regarding TAVI in small annuli but they focused on SEV devices only. Theyfound higher risk of PPM in case of intra annular devices and lower rates of PPM after postdilatation and valve oversizing.

Hase et al (Transcatheter aortic valve replacement with Evolut R versus Sapien 3 in Japanese patients with a small aortic annulus: The OCEAN-TAVI registry Catheterization and Cardiovascular Intervention, 2021) compared Sapiens 3 versus Corevalve and showed that Corevalve had better hemodynamic performance in small annuli. Device success and mortality was similar. Acurate valve were not included.

Actually, the question of small aortic annuli management in TAVI is burning. A randomized trial between Sapiens 3  and corevalve is launched (Hermann HC Am Heart J 2022) and will contribute to offer randomized data in this setting.

  1. The Novelty of these findings should be underscored if any, and the adjunctive value of the paper more clearly stated 

Thank you. Following your comment we now provide a conclusions section that emphasize our conclusions and its importance in daily practice.

In this study, we compared one BEV and two SEV valves in small aortic annuli. Few studies have explored the hemodynamic performances of different TAVI devices design in these anatomies. Our analysis is the first to compare the Sapiens 3 versus Corevalve versus Acurate in small aortic annuli treated with TAVI.

Our findings support the use of SEV TAVI valves in small anatomies. Indeed, SEV design was associated with lower rates of PPM.

CT scans should be studied in detail to identify patients at risk for PVL. Indeed, it seems that there is a compromise to do in small aortic annuli between PVL and PPM. BE may be favored in case of high risk of PVL (aortic annuli with important calcifications) while SE may be favored in case of high risk of PPM.

Special interest should be given to the choice of valve in patients with small aortic annuli undergoing a TAVR procedure.

Reviewer 2 Report

 well written and constructed paper.  Provides good information to the readers of the journal  about considerations for TAVR in patients with small aortic annuli.

The only downside to the paper is that the sample size is extremely small to truly determine 1 is better than the other in terms of the valves.

Author Response

This population is drawn from a monocentric patient cohort at La Timone University Hospital in Marseille. We share with you our experience on these small annuli and the strategy that we believe is the right one for these anatomies. Patients with a small aortic annulus seem to have a significant hemodynamic benefit on the occurrence of moderate or severe PPM with the use of an SE valve compared to a BE valve. Otherwise, CT scans should be studied in detail to identify patients at high risk for PVL. A different strategy could therefore be adopted in patients with a small aortic annulus by favouring BE valves when there is a high risk of PVL and SE valves when the risk of PPM is important.